# Co-Regulatory Roles of WC1 and WC2 in Asexual Development and Photoreactivation of *Beauveria bassiana*

**DOI:** 10.3390/jof9030290

**Published:** 2023-02-23

**Authors:** Si-Yuan Xu, Lei Yu, Xin-Cheng Luo, Sheng-Hua Ying, Ming-Guang Feng

**Affiliations:** Institute of Microbiology, College of Life Sciences, Zhejiang University, Hangzhou 310058, China

**Keywords:** entomopathogenic fungi, white collar proteins, aerial conidiation, solar UV resistance

## Abstract

The white collar proteins WC1 and WC2 interact with each other to form a white collar complex acting as a well-known transcription regulator required for the operation of the circadian clock in *Neurospora*, but their roles in insect-pathogenic fungal lifecycles remain poorly understood. Here, we report that WC1 and WC2 orthologs co-regulate the conidiation capacity and conidial resistance to solar ultraviolet-B (UVB) irradiation in *Beauveria bassiana*, after their high activities in the photorepair of UVB-induced DNA damages were elucidated previously in the insect mycopathogen, which features non-rhythmic conidiation and high conidiation capacity. The conidial yield, UVB resistance, and photoreactivation rate of UVB-impaired conidia were greatly reduced in the null mutants of *wc1* and *wc2* compared to their control strains. However, many other lifecycle-related phenotypes, except the antioxidant response, were rarely affected in the two mutants. Transcriptomic analysis revealed largely overlapping roles for WC1 and WC2 in regulating the fungal gene networks. Most of the differentially expressed genes identified from the null mutants of *wc1* (1380) and *wc2* (1001) were co-downregulated (536) or co-upregulated (256) at similar levels, including several co-downregulated genes required for aerial conidiation and DNA photorepair. These findings expand a molecular basis underlying the fungal adaptation to solar UV irradiation and offer a novel insight into the genome-wide co-regulatory roles of WC1 and WC2 in *B. bassiana*’s asexual development and in vivo photoreactivation against solar UV damage.

## 1. Introduction

Solar ultraviolet (UV) irradiation consisting of UVB (280–320 nm) and UVA (320–400 nm) wavelengths [1] is detrimental to formulated conidia as active ingredients of fungal insecticides [2,3,4,5], and hence restrains all-weather applications of fungal formulations [6,7,8]. UVB, a major solar UV component able to impair cellular macromolecules, is more detrimental to cells than UVA, which can induce the generation of intracellular reactive oxygen species [9,10]. Unveiling the molecular mechanisms behind insect-pathogenic fungal resistance to solar UV damage is crucial for the development and application of environmentally friendly fungal insecticides [8,11]. However, the understanding of such mechanisms remains limited.

DNA is a vital macromolecule vulnerable to UV irradiation, under which adjacent bases in DNA duplex are covalently linked to form cytotoxic photoproducts named cyclobutane pyrimidine dimer (CPD) and (6-4)-pyrimidine-pyrimidone (6-4PP) [12]. The DNA lesions of two adjacent nucleobases (typically thymidines) reduce cell viability or cause cell death [13,14,15]. Fungi have evolved two distinct UV-resisting mechanisms to recover from the CPD and 6-4PP lesions [16]. Photorepair is a rapid process of repairing shorter UV-induced DNA lesions under longer UV or visible light, which provides energy to break down the covalent linkages through the direct transfer of electrons to the CPD or 6-4PP lesions under the action(s) of photolyase(s) [17,18,19,20]. The photolyase– cryptochrome family (PCF) in filamentous fungi comprises no more than four members that share a similar DNA photolyase domain required for photorepair activity. Previous studies have revealed that fungal photorepair depends on only one or two photolyases (Phr1 specific to CPD and/or Phr2 specific to 6-4PP), rather than one or two DASH-type cryptochromes (Cry-DASHs) [21,22,23,24,25,26]. A reason for this dependence has long been unclear, until recently. Among three PCF members in *Beauveria bassiana*, an insect mycopathogen that is a main source of wide-spectrum fungal pesticides [27], Phr1 and Phr2 enable the photorepair of CPD and 6-4PP DNA lesions, respectively, and were proven to localize exclusively in the nucleus, whereas Cry-DASH (CryD), with no photorepair activity, was shown to localize only in the cytoplasm, despite its significant contribution to UVB resistance [26]. A more recent genome survey revealed the presence of a nuclear localization signal (NLS) motif in the amino acid sequences of Phr1 and Phr2, instead of Cry-DASH(s), in many fungi [8]. Particularly, Cry-DASH, as the sole PCF member in some mucoromycetous fungi, was reported to repair the CPD lesions in DNA in vitro or in vivo [28,29]. Indeed, the reported Cry-DASH was identical to fungal photolyases in both domain architecture and predictable NLS motif [8], although its subcellular feature was not disclosed in the previous studies. Thus, photorepair is a nucleus-specific cellular process required for the photoreactivation of UV-impaired fungal cells. The photoreactivation rate, i.e., recovery rate of UV-impaired cells to be easily assessed, is a reliable index of photorepair activity that can be assessed at a much higher cost.

Aside from photorepair, nucleotide excision repair (NER) has been well studied in the model yeast *Saccharomyces cerevisiae*. Unlike photorepair, which depends on one or two photolyases, however, NER relies on manifold enzymes and proteins involved in proteasome activity and poly-ubiquitination in the dark, including helicases, endonucleases, polymerases and ligases [16,30,31,32,33]. In the yeast, early studies resulted in the identification of anti-UV radiation (RAD) genes encoding a large family of RAD proteins and partners [34,35]. Those proteins form multiple RAD–RAD and RAD-containing complexes that function in the NER pathway to regulate the recognition, opening, incision, and/or repair of UV-induced DNA lesions [36]. As examples, the Rad1–Rad10 complex formed through an interaction between Rad1 and Rad10 acts as an endonuclease enabling the recognition of the junction of single- and double-strand DNAs and removing unpaired 3′ tails from the junction [37,38,39,40,41]. The Rad4–Rad23–Rad33 complex, formed through interactions of Rad4 with Rad23 and Rad33, can sense a distorted DNA duplex to initiate global-genome NER (GG-NER) by its recruitment to impaired DNA sites through interactions of Rad4 with chromatin remodeling complexes [31,42,43]. However, it is unclear whether the principles for the yeast NER are applicable to filamentous fungi, in which NER has been rarely studied.

Solar UV is close to blue light in wavelength. WC1 and WC2 are white collar proteins that interact with each other to form a heterodimeric white collar complex (WCC), which acts as a well-known blue-light sensor and a regulator of the circadian clock in *Neurospora crassa* [44,45,46,47,48,49] and the insect mycopathogen *Metarhizium robertsii* [50]. Aside from a core role in the clock, WCC regulates hundreds of light-responsive genes, including the coding genes of many transcription factors and enzymes/proteins involved in post-translational modifications and chromatin remodeling [19,20]. Previously, the expression of *phr1* was light-induced by *blr-1* (*wc1*) and *blr-2* (*wc2*) in *Trichoderma atroviride* [51], and its promoter featured an identified light-responsive region comprising typical WCC binding sites [22]. The photoreactivation rate of UV-impaired cells was lowered in the absence of *wco1* (*wc1*) as much as in the absence of both *phr1* and *phr2* in *Ustilago maydis* [23]. While the previous study [23] suggested the requirement of WC1 for fungal photoreactivation, it is obscure why WC1 lacking a DNA photolyase domain had the photoreactivation activity reliant on photorepair. It is also obscure whether WC2 is involved in the transcriptional activation of photolyase genes required for photorepair. These puzzles have been partially solved in recent studies. In *M. robertsii*, WC1 and WC2 were proven to interact with each other and also with both Phr1 and Phr2, resulting in the abolished expression of *phr2* and *phr1* and thenull photorepair of 6-4PP and CPD DNA lesions in the null mutants of *wc1* and *wc2* [52], respectively. In *B. bassiana*, either WC1 or WC2 was shown to act as a regulator of both *phr1* and *phr2* and photorepair the CPD and 6-4PP DNA lesions more efficiently than either photolyase alone [53]. Moreover, the orthologs of Rad1 and Rad10 that form the Rad1–Rad10 complex required for the yeast GG-NER [37,38,39,40,41] were shown to have acquired extraordinarily high photoreactivation activities through their interactions with WC1 and WC2 in *B. bassiana* [53] and with Phr1 and WC2 in *M. robertsii* [54]. The homologs of Rad4 and Rad23 essential for the yeast GG-NER [31,42,43] were also proven to interact with each other and have photoreactivation activities due to interactions of Rad23 with Phr2 and WC2 in *B. bassiana* [55,56]. More interestingly, the mentioned RAD orthologs or homologs did show an NER activity that was observable only after a dark incubation exceeding 24 h, which is virtually infeasible for the fungal insect pathogens on the Earth’s surface. The recent advances demonstrate much more complicated mechanisms underlying photorepair-reliant photoreactivation, which had been long considered to depend on only one or two photolyases, and confirm a hypothesis that photorepair may serve as a uniquely feasible mechanism behind filamentous fungal adaptation to solar UV and is likely regulated by the WCC-cored pathway comprising not only photolyases but many more anti-UV RAD proteins [8]. While the roles of WC1 and WC2 in the transcriptional activation of Phr1 and Phr2 have become clear, a genome-wide view of their regulatory roles is still absent in the fungal insect pathogens. This study seeks to elucidate possible roles of WC1 and WC2 in the in vitro and in vivo lifecycles of *B. bassiana* through phenotypic and transcriptomic analyses of their null mutants. Our goal is to gain an in-depth insight into the regulatory roles of WC1 and WC2 in the fungal adaptation to its insect host and environment. As presented below, our data demonstrate the essential roles for WC1 and WC2 in mediating asexual development and the cellular response to solar UV irradiation.

## 2. Materials and Methods

### 2.1. Fungal Strains and Media

The deletion mutants (DM) and complementation mutants (CM) of *wc1* (tag locus: BBA_10271) and *wc2* (BBA_01403) generated previously [53] in the background of the wild-type strain *B. bassiana* ARSEF 2860 (designated WT) were used in all experiments, with each including three independent replicates. Their colonies were maintained on Sabouraud dexgtrose agar (4% glucose, 1% peptone, and 1.5% agar) plus 1% yeast extract (SDAY) or ¼ SDAY (amended with one quarter of each SDAY nutrient). The minimal medium Czapek-Dox agar (CDA; 3% sucrose, 0.3% NaNO_3_, 0.1% K_2_HPO_4_, 0.05% KCl, 0.05% MgSO_4_, 0.001% FeSO_4_, and 1.5% agar) was used in stress assays. A germination medium (GM; 2% sucrose, 0.5% peptone, and 1.5% agar) was used in the assays for conidial UVB resistance and activities in photoreactivation and dark reactivation of UVB-impaired conidia.

### 2.2. Assays for Radial Growth Rates under Normal Conditions and Different Stresses

The DM and control (WT and CM) strains were grown on SDAY, ¼ SDAY, and CDA by spotting 1 μL of a 10^6^ conidia/mL suspension per plate. After a 7-day incubation at the optimal regime of 25 °C and 12:12 (L:D), the diameter of each colony was assessed as a growth index using two measurements perpendicular to each other across its center.

To assay each strain’s responses to different types of stress cues, its radial growth was initiated as above on CDA plates alone (control) or supplemented with H_2_O_2_ (2 or 4 mM) or menadione (0.02 mM) for oxidative stress, NaCl (0.4 M), KCl (0.4 M) or sorbitol (1 M) for osmotic stress, Congo red (3 μg/mL) or calcofluor white (10 μg/mL) for cell-wall stress, and methyl methanesulfonate (25 μg/mL) or hydroxyurea (5 mM) for DNA perturbing stress. The diameter of each colony was estimated as above after a 7-day incubation at 25 °C. Additionally, 2-day-old SDAY colonies initiated at 25 °C were exposed to heat shock at 42 °C for 3, 6 or, 9 h, followed by 5-day growth recovery at 25 °C and estimation of colony diameters as above. The diameter measurements of stressed colonies (*d*_s_) and control colonies (*d*_c_) were used to calculate the relative growth inhibition (RGI = (*d*_c_ − *d*_s_)/*d*_c_ × 100) of each strain as an index of its sensitivity to each stress.

### 2.3. Assessment of Conidiation Capacity

SDAY cultures were initiated by spreading 100 μL of a 10^7^ conidia/mL suspension evenly on each plate (9 cm diameter), and incubated for 9 days at the optimal regime. From day 5 onwards, three plugs were taken every 2 days from each plate culture using a cork borer (5 mm diameter). Conidia on each plug were released into 1 mL of 0.05% Tween 80 through 10 min supersonic vibration, followed by microscopic assessment of conidial concentration in the suspension with a Neubauer hemocytometer and conversion of the concentration to the number of conidia per square centimeter of plate culture.

### 2.4. Bioassays for Fungal Virulence

The virulence of each strain to *Galleria mellonella* larvae (4th instar) was assayed in two infection modes. Normal cuticle infection (NCI) was initiated by immersing three groups of ~35 larvae for 10 s in 40 mL aliquots of a 10^7^ conidia/mL suspension. Cuticle-bypassing infection (CBI) was initiated by injecting 5 μL of a 10^5^ conidia/mL suspension into the hemocoel of each larva in each group using a microinjector. All groups of larvae infected in either mode were held at 25 °C for survival/mortality records every 12 or 24 h until no more change in mortality occurred. Median lethal time (LT_50_) was estimated as an index of virulence to the model insect by analyzing the resultant time–mortality trend in each group of larvae infected per strain in either mode.

### 2.5. Assays for Conidial Resistance to UVB Irradiation

Conidial UVB resistance of each strain was assayed in a Bio-Sun^++^ UV irradiation chamber (Vilber Lourmat, Marne-la-Vallée, France) as described previously [43]. For each strain, 100 μL aliquots of a 10^7^ conidia/mL suspension were spread onto GM plates. After 10 min air drying of sterile ventilation, the plates were exposed to UVB irradiation at the gradient doses of 0.025 to 0.5 J/cm^2^ (three plates per dose) in the sample tray of the chamber, where the wavelength (weighted 312 nm) and intensity of irradiation were automatically adjusted four times per second by an inset microprocessor to control an error of ≤1 μJ/cm^2^ (10^–6^) for the irradiation at a given dose (following the manufacturer’s guide). Upon irradiation, the plates were covered with lids and incubated for 24 h in full darkness at 25 °C. Three plates not exposed to UVB irradiation were used as a control for each strain. From 8 h incubation onwards, conidial germination percentage was assessed every 2 h from three microscopic view fields (100× magnification) per plate until the maximum appeared after exposure to each dose. Survival index (*I*_s_) was computed as a ratio of maximal germination percentages of irradiated versus non-irradiated conidia with respect to the control. The observed *I*_s_ trend over the gradient doses (*d*) was fitted to the modified logistic equation *I*_s_ = 1/[1 + exp(*a* + *rd*)] [43]. The fitted parameters *a* and *r* (dose-dependent declining rate of conidial survival) were used to estimate lethal doses (LD*_x_*) as an index of conidial UVB resistance, where *x* denotes a percentage of UVB-inactivated conidia (*x* = (1 −*I*_s_) × 100 at the *I*_s_ value of 0.75, 0.5, 0.25 or 0.05).

### 2.6. Assays for Photoreactivation and Dark Reactivation Rates

For each DM or control strain, GM plates smeared evenly with 100 μL aliquots of a 10^7^ conidia/mL suspension were exposed to UVB doses of 0.2, 0.3, and 0.4 J/cm^2^ in the UV chamber. The irradiated plates were incubated immediately at 25 °C for 5 h under white light and then 19 h in the dark (photoreactivation via photorepair) or directly for 24 h in the dark (dark reactivation via NER). Maximal germination percentages of conidia severely impaired or inactivated at the stated UVB doses were monitored as aforementioned during the period of dark incubation after the first 5 h of light or dark incubation.

### 2.7. Transcriptomic Analysis

The Δ*wc1*, Δ*wc2*, and WT cultures were initiated by spreading 100 μL aliquots of a 10^7^ conidia/mL suspension on cellophane-overlaid SDAY plates and incubated for 3 days at the optimal regime. Three cultures (replicates) per strain were sent to Lianchuan BioTech (Hangzhou, China) for generation of transcriptome. All clean tags from RNA- seq datasets were mapped to the *B. bassiana* genome [57]. Differentially expressed genes (DEGs) were identified at significant levels of both the log_2_ ratio (fold change) ≤−1 or ≥1 and *q* < 0.05, and enriched to three function categories (*p* < 0.05) through gene ontology (GO) analysis at http://www.geneontology.org (accessed on 21 February 2023), and pathways (*p* < 0.05) through Kyoto Encyclopedia of Genes and Genomes (KEGG) analysis at http://www.genome.jp/kegg (accessed on 21 February 2023).

### 2.8. Statistical Analysis

One-factor (fungal strains) analysis of variance was carried out to differentiate between data from each of the experiments including three independent replicates, followed by multiple comparison of mean parameters through Tukey’s test.

## 3. Results

### 3.1. WC1 and WC2 Are Required for Conidiation and Conidial UVB Resistance

The 7-day-old colonies of both *wc1* DM and *wc2* DM grown at the optimal regime after inoculation with ~10^3^ conidia showed a whitish fluffy phenotype on the rich medium SDAY or 1/4 SDAY (Figure 1a), but no change in colony diameter (Figure 1b) in comparison to the control strains. In the assays for responses to different types of chemical stressors added to CDA plates, only the *wc2* DM displayed a significant increase (*p* < 0.05 in Tukey’s test) in sensitivity to oxidative stress induced by H_2_O_2_ (2 or 4 mM) or menadione (0.02 mM) (Figure 1c). However, the DM strains of *wc1* and *wc2* and their control strains were equally responsive to osmotic stress induced by NaCl, KCl, or sorbitol, cell-wall stress induced by Congo red or calcofluor white, and DNA-perturbing stress induced by methyl methanesulfonate or hydroxyurea. Growth recovery rates at 25 °C did not differ significantly between the DM and control strains after 2-day-old SDAY colonies were exposed to 42 °C heat shock for 3, 6, or 9 h.

The mean (±SD, *n* = 9) conidial yields of the control strains in the 5-, 7- and 9-day-old cultures initiated by spreading 100 μL aliquots of a 10^7^ conidia/mL suspension and incubated at the optimal regime were measured as 28.6 (±3.5) × 10^7^, 70.0 (±3.3) × 10^7^ and 77.6 (±6.6) × 10^7^ conidia/cm^2^,respectively (Figure 1d). In contrast, the Δ*wc1* and Δ*wc2* mutants’ conidial yields were not measurable on Day 5 and sharply reduced by 98.1% and 98.5% on Day 7, respectively, in comparison to the control strains’ mean yield. The yield reduction on Day 9 decreased to 63.6% for Δ*wc1* and 73.9% for Δ*wc2*. These data demonstrated a marked delay of conidiation and a large reduction of conidiation capacity in the absence of *wc1* or *wc2*. In the standardized bioassays, the estimates of LT_50_ as an index of virulence showed no variation among the DM and control strains tested via NCI (*F*_4,10_ = 0.87, *p* = 0.51) or CBI (*F*_4,10_ = 2.39, *p* = 0.12), as illustrated in Figure 1e, indicating no role of either WC1 or WC2 in the fungal infection cycle.

In the assays for conidial UVB resistance by 24-h dark incubation of irradiated conidia at 25 °C, conidial survival trends over the gradient UVB doses of 0.025–0.5 J/cm^2^ (Figure 1f) fitted well with the modified logistic equation (r^2^ ≥ 0.98, *p* < 0.001 for fitness *F* test). The averages for control strains’ LD_25_, LD_50_, LD_75_, and LD_95_ (*n* = 9) estimated from the fitted equations were 0.169 (±0.011), 0.223 (±0.009), 0.278 (±0.009), and 0.369 (±0.014) J/cm^2^, respectively (Figure 1g). Compared to these means, the Δ*wc1* and Δ*wc2* mutants’ LD_25_, LD_50_, LD_75_, and LD_95_ were reduced by 67% and 74%, 63% and 68%, 60% and 65%, and 57% and 61%, respectively. These reductions indicated that the two mutants were severely compromised in conidial UVB resistance, which was ~5% more impaired in Δ*wc2*.

Altogether, both WC1 and WC2 played essential roles in the asexual development and conidial UVB resistance of *B. bassiana*, although WC2 contributed slightly more to the two traits. An increase in the Δ*wc2* mutant’s sensitivity to oxidative stress implicated an involvement of WC2 in the fungal antioxidant response. However, both of them were dispensable for radial growth under normal culture conditions, cellular responses to osmotic, cell-wall perturbing, DNA perturbing, and heat-shocking stresses, and cellular processes and events associated with the fungal insect pathogenicity and virulence.

### 3.2. Roles of WC1 and WC2 in Photoreactivation and Dark Reactivation

Previously, WC1 or WC2 was confirmed as a regulator of both *phr1* and *phr2*, whose expression was nearly abolished in the Δ*wc1* and Δ*wc2* mutants severely compromised in the photorepair of UVB-induced CPD and 6-4PP DNA lesions [53]. The activity of each in photoreactivation and dark reactivation of conidia differentially impaired at the UVB doses of 0.3, 0.4, and 0.5 J/cm^2^ was assessed using an incubation of 5 h under white light plus 19 h in the dark and a 24-h dark incubation at 25 °C. The control strains had many more impaired conidia reactivated than the DM strains at the end of dark incubation after a 5-h light exposure (Figure 2a). Direct 24-h dark incubation also reactivated some conidia of the control strains instead of the DM strains, whose impaired conidia were not reactivated in the dark. In contrast, non-irradiated conidia of all strains were reactivated using a 12-h incubation regardless of the light exposure (Figure 2b).

The reactivated percentage of the control strains’ conidia was dependent on the UVB dose and incubation time post-irradiation. In the full dark treatment, the control strains’ conidia impaired at 0.3 J/cm^2^ were on average reactivated by 2% (±1.4), 17% (±1.4), and 30% (±2.3) (*n* = 9) at the end of 12, 18, and 24 h incubation, respectively (Figure 2c). When the UVB dose increased to 0.4 and 0.5 J/cm^2^, the reactivated percentage of the control strains at the end of 24 h dark incubation decreased to 10.2% (±2.0) and 1.3% (±0.5), respectively. Although it was not observable at the UVB dose of 0.3 J/cm^2^, 78% and 60% of the Δ*wc1* and Δ*wc2* mutants’ conidia impaired at 0.05 J/cm^2^ were reactivated through 24 h dark incubation and the reactivated percentages dropped to only 2% and 1% at 0.2 J/cm^2^, respectively (shown in Figure 1f). These data demonstrated that the NER-dependent reactivation activity was greatly reduced in the absence of *wc1* or *wc2*. However, the NER activity was not enough to reactivate the severely impaired conidia using 12 h dark incubation even in the presence of *wc1* and *wc2*, implying an infeasibility for the dark reactivation in the field where the night-time (dark) is too short for NER.

In the photoreactivation treatment, the control strains’ conidia severely impaired at 0.3 J/cm^2^ were increasingly reactivated by the first 5-h light exposure to 21% (±3.2), 75% (±2.8), and 98% (±1.0) at the end of 7, 13, and 19 h dark incubation, respectively (Figure 2d). The photoreactivation rates (*n* = 9) of their conidia inactivated at 0.4 and 0.5 J/cm^2^ were 70% (±2.6) and 45% (±1.4) at the end of 13 h dark incubation and 95% (±2.7) and 83% (±3.0) at the end of 19 h dark incubation, respectively. In contrast, the photoreactivation rates of the Δ*wc1* and Δ*wc2* mutants’ conidia at the end of 19 h dark incubation were drastically lowered to ~14% and ~4% after exposure to 0.3 and 0.4 J/cm^2^,respectively, and were undetectable at 0.5 J/cm^2^. These observations highlighted essential roles for WC1 and WC2 in photoreactivation in vivo to prevent *B. bassiana* from suffering UVB damage.

### 3.3. Genome-Wide Insight into Overlapped Regulatory Roles of WC1 and WC2

The genome-wide regulatory roles of WC1 and WC2 were analyzed using transcriptomes constructed with three independent cultures (replicates) of the Δ*wc1*, Δ*wc2*, and WT strains incubated for 3 days at the optimal regime. The transcriptomes contained 1380 DEGs (up/down ratio: 481:899; the same meaning for ratios mentioned below) from Δ*wc1* (Figure 3a, Appendix A) and 1001 DEGs (360:641) from Δ*wc2* (Figure 3b, Appendix A). The majority of those DEGs (log_2_ ratio ≥1 or ≤−1, *q* < 0.05) were co-upregulated (256) or co-downregulated (536) in the Δ*wc1* and Δ*wc2* mutants (Figure 3c), indicating that the gene networks regulated by WC1 and WC2 largely overlapped in *B. bassiana*.

In the GO analysis, 1631 and 1197 DEGs (531:1100 and 392:805) of the Δ*wc1* and Δ*wc2* mutants were enriched to 108 and 96 GO terms of three function categories at the significant level of *p* < 0.05 (Appendix A), respectively. Up to 61 GO terms (Appendix A) were dominated by co-downregulated genes in the two mutants (main terms shown in Figure 3d); most of the remaining GO terms enriched in Δ*wc1* or Δ*wc2* alone contained only a few DEGs per capita. The GO analysis reinforced the largely overlapping roles of WC1 and WC2 in mediating the gene networks involved in the cellular component, biological process, and molecular function.

In the KEGG analysis, 20 KEGG pathways were enriched with 278 DEGs (92:186) from Δ*wc1* (Appendix A) and 220 DEGs (71:149) from Δ*wc2* (Appendix A) at a significance level of *p* < 0.05. The enriched pathways were all dominated by downregulated genes and mostly co-downregulated in the two mutants (Figure 3e). Aside from these pathways, four other pathways were downregulated in Δ*wc1* or Δ*wc2* alone. Almost all enriched pathways were involved in carbon and nitrogen metabolism. Exceptionally, one co-downregulated pathway (ID: map02010) was involved in environmental information processing (membrane transport: ABC transporters). The other pathway (ID: map04146), enriched to the Δ*wc2* mutant alone, was involved in the cellular process (transport and catabolism: peroxisome). The up/down ratio of DEGs enriched to the peroxisome pathway was 4:9, coinciding with the increased sensitivity of the Δ*wc2* mutant to oxidative stress. The KEGG analysis again reinforced the largely overlapping roles for WC1 and WC2 in the genome-wide transcription regulation of *B. bassiana*.

### 3.4. Regulatory Roles of WC1 and WC2 in Asexual Development and Photoreactivation

For deeper insight into conidiation capacity and photoreactivation activity severely compromised in the DM strains of *wc1* and *wc2*, several related genes functionally characterized in *B. bassiana* were found in the transcriptomes and are listed in Table 1.

Among the four genes required for conidiation and conidial maturation [58,59], the key gene *brlA* to activate the central developmental pathway (CDP) was largely downregulated in both Δ*wc1* and Δ*wc2*. Two frequency genes (*frq1* and *frq2*), which are involved in the activation of CDP and are required for non-rhythmic conidiation in response to photoperiod change [60], were significantly downregulated in Δ*wc2*, but only *frq1* was downregulated in Δ*wc1*. The expression of *vvd*, which encodes another blue-light photoreceptor also involved in the activation of CDP and is required for conidiation and responding to the photoperiod [61], was abolished in Δ*wc1* (log_2_ ratio = −4.60) and Δ*wc2* (log_2_ ratio = −5.85). The reduced expression levels of these conidiation-required genes coincided well with the two mutants’ severe defects in aerial conidiation. For the Δ*wc2* mutant, the downregulation of both *frq1* and *frq2*, and more reduced *vvd* expression level, were also in agreement with its conidiation capacity being more compromised.

The expression of *wc2* was significantly upregulated (log_2_ ratio = 1.07) in the absence of *wc1* but not vice versa. Aside from *wc1* and *wc2* as regulators of photolyases [53], the created transcriptomes were further analyzed to reveal expression levels of several genes required for photorepair, photoreactivation, and UV resistance. Among three PCF genes, *phr1* was more downregulated in Δ*wc2* (log_2_ ratio = −4.40) than in Δ*wc1* (log_2_ ratio = −3.64), while *phr2* was downregulated at similar, but insignificant, levels in both mutants. In addition, *cryD*, which contributes to conidial UVB resistance (as does *phr2* but it has no photorepair activity [26]), was also markedly co-downregulated in the two mutants. Previously, the anti-UV proteins Rad1, Rad10, Rad23, and Rad4A were shown to have high photoreactivation activities due to direct or indirect links of each to Phr1, Phr2, and/or regulators of Phr1 and Phr2 [53,55,56]. In this study, however, the expression levels of their coding genes in the two mutants were not affected at the significant levels, suggesting that those links relying on multiple protein–protein interactions occur as post-translational events. Obviously, the Δ*wc1* and Δ*wc2* mutants’ severe defects in photoreactivation were ascribed to differential downregulation of their PCF genes.

## 4. Discussion

Our experimental data demonstrate that WC1 and WC2 play essential, but similar, roles in mediating both photoreactivation and asexual development in *B. bassiana*. WC2 was also shown to take part in the cellular response to oxidative stress. Such roles were all similar to those of their orthologs in *M. robertsii* [52]. However, both WC1 and WC2 were not involved in cellular processes and events associated with the insect-pathogenic lifecycle of *B. bassiana*. This is different from significant contributions of their orthologs to the virulence of *M. robertsii* via normal cuticle infection.

The activities of WC1 and WC2 In the photorepair of UVB-induced CPD and 6-4PP DNA lesions were previously shown to elucidate the high photoreactivation activities of Rad1 and Rad10 in the two insect mycopathogens [52,53,54]. Interestingly, the links of WC1 and WC2 to Phr1, Phr2, Rad1, and Rad10 are somewhat different between the two fungi. In *B. bassiana*, either WC1 or WC2 was proven to act as a regulator of both *phr1* and *phr2* by the activity of its binding to the promoter region of each photolyase gene and interact with both Rad1 and Rad10, but not Phr1 or Phr2 [53]. In *M. robertsii*, WC1 was shown to interact with both Phr1 and Phr2, as does WC2 [52], but not interact with Rad1 or Rad10, while only WC2 was able to interact with Rad1 linked to Phr1 by the Rad1–Rad10 and Phr1–Rad10 interactions [54]. Despite the subtle differences, the photoreactivation activities of either WC1 and WC2 or Rad1 and Rad10 were similar in the two fungi, suggesting core roles for WC1 and WC2 in the insect-pathogenic fungal adaptation to solar UVB. Notably, conidial UVB resistance indicated by LD_50_ was much more compromised in the present Δ*wc1* (63% decrease) and Δ*wc2* (68%) mutants than in the previous Δ*phr1* (38%) and Δ*phr2* (19%) mutants [26]. The conidia inactivated at the UVB dose of 0.5 J/cm^2^ were photoreactivated by 7% and 34% in the previous Δ*phr1* and Δ*phr2* mutants [26], but not at all in the present Δ*wc1* and Δ*wc2* mutants. These findings reinforced the much greater roles of WC1 and WC2 than of Phr1 and Phr2 in protecting *B. bassiana* from solar UV damage and also support the previous report on an anti-UV role of WC1 in *U. maydis* [23]. The regulatory roles for WC1 and WC2 in the transcriptional activation of anti-UV PCF genes were confirmed in the analyzed transcriptomes, in which both *phr1* and *cryD* were co-downregulated at high levels. Exceptionally, the expression of *phr2* was nearly abolished in the Δ*wc1* and Δ*wc2* mutants previously analyzed using real-time quantitative PCR [53], but was downregulated insignificantly at similar levels in the present transcriptomes. This suggests a bias between the two methods used for transcriptional profiling. In addition, the coding genes of Rad1, Rad10, Rad4A, and Rad23 essential for fungal photoreactivation [53,55,56] were not differentially regulated in the two mutants. This suggests that the direct/indirect links of such anti-UV RAD proteins to WC1, WC2, Phr1, and/or Phr2 occur at the post-translational level. The GO term methylation was enriched with multiple DEGs from the Δ*wc1* and Δ*wc2* mutants, implicating an involvement of either WC1 or WC2 in methylation as a crucial post-translational event in the nucleus.

Moreover, our transcriptomic analysis confirmed overlapping roles for WC1 and WC2 in mediating the asexual lifecycle of *B. bassiana*, which usually lacks a teleomorph. The WT strain used in this study featured non-rhythmic conidiation controlled by Frq1 and Frq2, which were proven to have opposite time-course nuclear dynamics that orchestrate expressions of the CDP activator genes *brlA*, *abaA*, and *wetA* consistently in day-time and night-time to achieve conidiation capacity in approximately one week [60]. The fungal conidiation was abolished in the absence of *brlA* or *abaA* [58] and severely compromised in the absence of *frq1* or *frq2* [60] and by the deletion of *vvd*, a coding gene of another blue-light photoreceptor that is nucleocytoplasmic shuttling in response to photoperiod change [61]. In the transcriptomes, *brlA* was largely repressed in the Δ*wc1* and Δ*wc2* mutants, accompanied by the suppressed expression of *frq1* in the two mutants and downregulation of *frq2* in Δ*wc2*. Particularly, the *vvd* expression was repressed to a hardly detectable level in the two mutants. These results indicate overlapping roles for WC1 and WC2 in transcriptional activation of those genes required for the asexual development of *B. bassiana* and offer an answer to why the fungal conidiation capacity was severely compromised in the absence of *wc1* or *wc2*.

In conclusion, our RNA-seq datasets shed light upon overlapping roles for WC1 and WC2 in co-regulating the asexual cycle in vitro, the photoreactivation of UVB-impaired conidia in *B. bassiana*, and an involvement of WC2 in the peroxisome pathway that is important for fungal resistance to oxidative stress. However, the datasets provide little clue to the high photoreactivation activities of Rad1, Rad10, Rad4A, and Rad23, which were shown to be directly or indirectly linked to WC1, WC2, Phr1, and/or Phr2 in previous studies [26,53,55,56]. Therefore, the direct/indirect links through multiple protein–protein interactions elucidated in previous studies were inferred to be post-translational nuclear events. Both the datasets and the inference implicate that WC1 and WC2 act as core regulators at both transcriptional and post-translational levels to support *B. bassiana*’s resistance and adaptation to solar UV irradiation.

## Figures and Tables

**Figure 1 jof-09-00290-f001:**
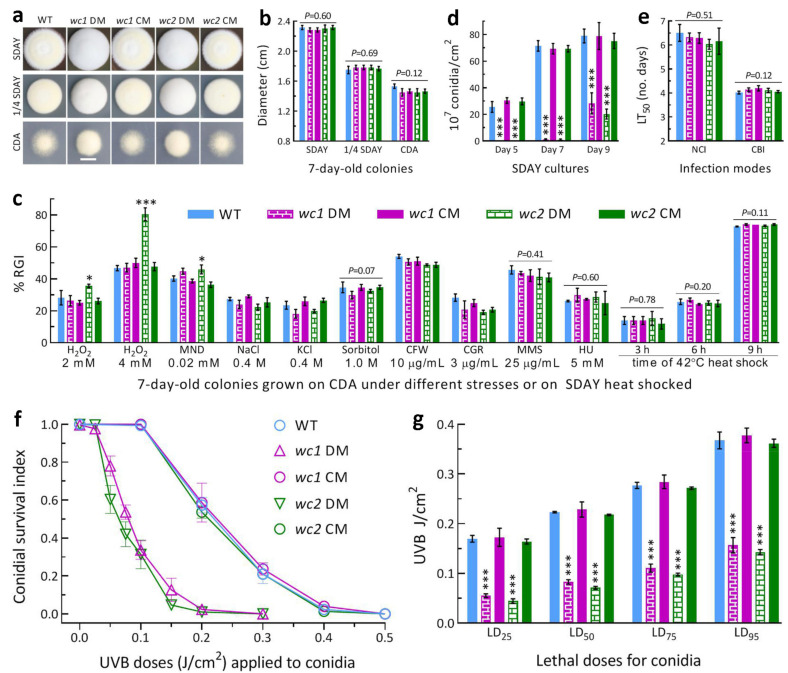
Roles of WC1 and WC2 in the lifecycles in vitro and in vivo of *B. bassiana*. (**a**,**b**) Colony images (scale: 10 mm) and diameters of fungal strains (WT, wild-type; DM, disruption mutant; CM, complemented mutant) incubated on SDAY, 1/4 SDAY, and CDA for 7 days at the optimal regime of 25 °C and 12:12 (L:D). (**c**) Relative growth inhibition (RGI) of fungal colonies incubated at 25 °C for 7 days on CDA supplemented with indicated concentrations of different chemicals (MND, menadione; CFW, calcofluor white; CGR, Congo red; MMS, methyl methanesulfonate; HU, hydroxyurea) and on SDAY for 5-day growth recovery after 2-day-old colonies were exposed to 42 °C for 3, 6, and 9 h of heat shock. Each colony was initiated with ~10^3^ conidia. (**d**) Conidial yields in the SDAY cultures incubated at the optimal regimes for 5, 7, and 9 days after initiation by spreading 100 μL aliquots of a 10^7^ conidia/mL suspension. (**e**) LT_50_ for the virulence of each strain against *G. mellonella* larvae via normal cuticle infection (NCI) and cuticle-bypassing infection (CBI). (**f**,**g**) Conidia survival trends over the gradient UVB doses and lethal doses (LD_x_) estimated from the trends using modeling analysis. *p* < 0.05 * or 0.001 *** in Tukey’s test. Error bars: standard deviations (SDs) of the means from three independent replicates.

**Figure 2 jof-09-00290-f002:**
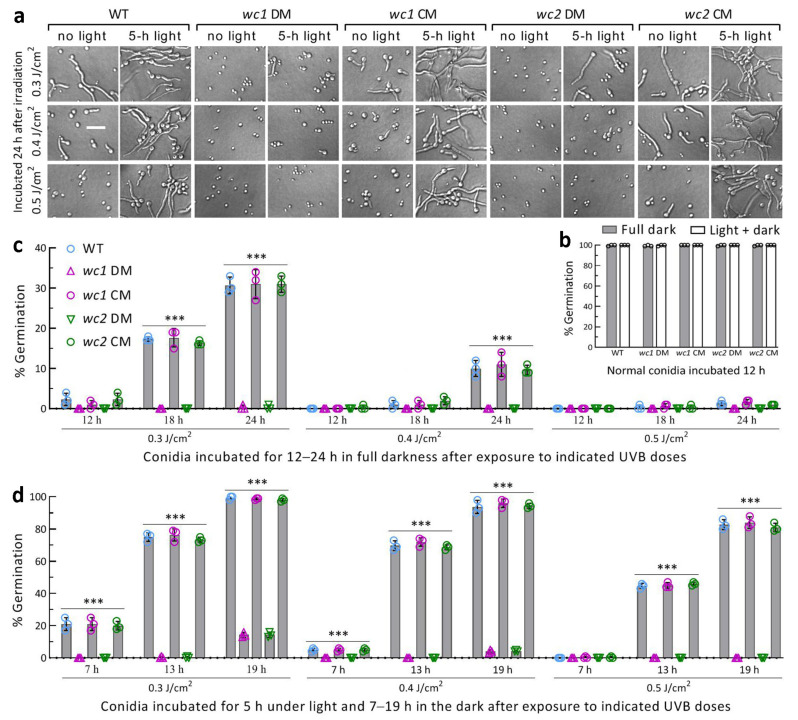
Essential versus infeasible roles of WC1 and WC2 in photoreactivation versus dark reactivation of UVB-impaired *B. bassiana* conidia. (**a**) Microscopic images (scale: 20 μm) for conidial germination status of disruption mutants (DM) and control (CM and WT) strains incubated at 25 °C for 5 h under white light plus 19 h in the dark (photoreactivation),or directly for 24 h in the dark (NER) after irradiation at the indicated UVB doses. (**b**) Germination percentages of normal conidia incubated at 25 °C for 5 h under the light plus 7 h in the dark and directly for 12 h in the dark. (**c**,**d**) Germination percentages of irradiated conidia reactivated using 12, 18, and 24 h of full dark incubation and the incubation of 5 h light plus 7, 13, and 19 h in the dark at 25 °C, respectively. *** *p* < 0.0001 in Tukey’s test. Error bars: SDs of the means from three independent replicates.

**Figure 3 jof-09-00290-f003:**
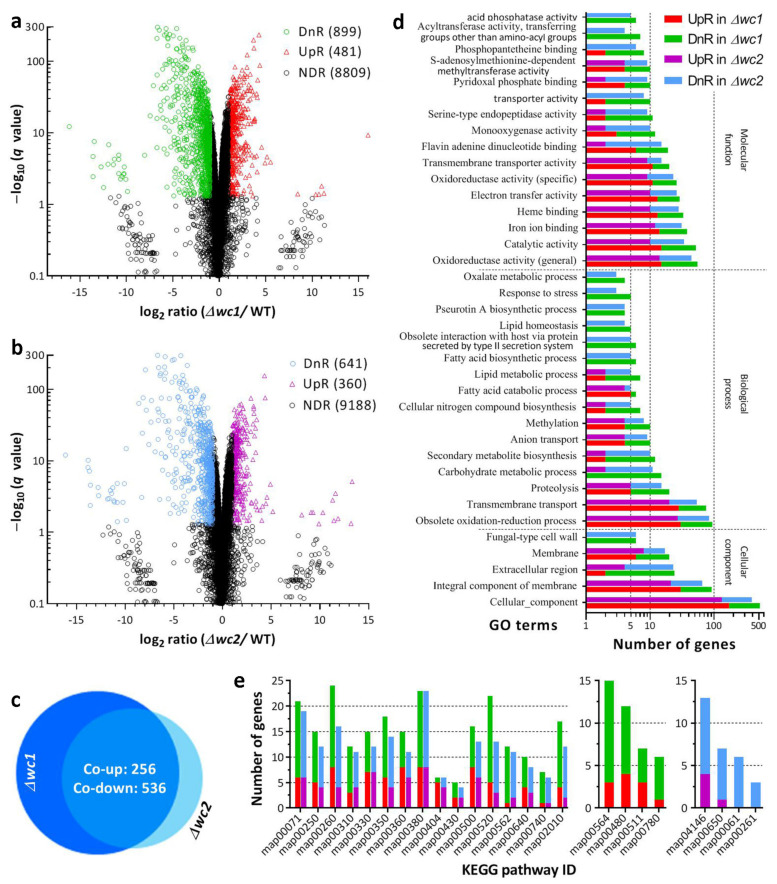
Overlapping roles of WC1 and WC2 in regulating gene expression networks of *B. bassiana*. (**a**,**b**) Distributions of log_2_ ratio values and *q* values for all genes in the transcriptomes constructed with three 3-day-old SDAY cultures (replicates) of the Δ*wc1*, Δ*wc2*, and WT strains grown at the optimal regime of 25 °C and L:D 12:12. DnR and UpR denote downregulated (log_2_ ratio ≤ −1) and upregulated (log_2_ ratio ≥ 1) genes at the significant level of *q* < 0.05, respectively. NDR, not differentially regulated at significant levels. (**c**) Counts of genes significantly co-upregulated and co-downregulated in Δ*wc1* and Δ*wc2*. (**d**) Counts of differentially expressed genes (DEGs) enriched (*p* < 0.05) to main GO terms co-downregulated in Δ*wc1* and Δ*wc2*. (**e**) Counts of DEGs enriched (*p* < 0.05) to KEGG pathways co-downregulated in both Δ*wc1* and Δ*wc2* (left), and downregulated in Δ*wc1*(middle) or Δ*wc2* (right) alone.

**Table 1 jof-09-00290-t001:** The log_2_ ratio values of crucial phenotype-related genes in the analyzed transcriptomes.

Gene	Tag Locus	Description	log_2_Ratio *		Ref.
			Δ*wc1*/WT	Δ*wc2*/WT	
Involved in aerial conidiation and conidial maturation					
*brlA*	BBA_07544	CDP activator BrlA	−2.595 *	−1.971 *	[58]
*abaA*	BBA_00300	CDP activator AbaA	−0.291	−0.190	[58]
*wetA*	BBA_06126	CDP activator WetA	−0.620	−0.405	[59]
*vosA*	BBA_01023	Velvety protein VosA	0.921	0.793	[59]
Involved in non-rhythmic conidiation inresponse to photoperiod					
*frq1*	BBA_01528	Frequency clock protein 1	−1.131 *	−1.524 *	[60]
*frq2*	BBA_08957	Frequency clock protein 2	−0.648	−1.313 *	[60]
*vvd*	BBA_02876	Blue-light photoreceptor VVD	−4.600 *	−5.851 *	[61]
Involved in UVB resistance, photorepair, and photoreactivation					
*wc1*	BBA_10271	White collar protein 1 WC1	−6.636 *	−0.030	[53]
*wc2*	BBA_01403	White collar protein 2 WC2	1.066 *	−4.293 *	[53]
*phr1*	BBA_01664	CPD-specific DNA photolyase	−3.637 *	−4.404 *	[26]
*phr2*	BBA_01034	6-4PP-specific DNA photolyase	−0.662	−0.640	[26]
*cryD*	BBA_02424	DASH-type cryptochrome	−1.872 *	−1.968 *	[26]
*rad1*	BBA_07749	Photoreactivation-required Rad1	0.739	0.386	[53]
*rad10*	BBA_03417	Photoreactivation-required Rad10	−0.098	0.032	[53]
*rad23*	BBA_01030	Photoreactivation-required Rad23	0.296	0.077	[55]
*rad4A*	BBA_02814	Photoreactivation-required Rad4A	0.390	0.166	[56]

* Table entries are significant at *q* < 0.05.

## Data Availability

All experimental data are included in this paper and Appendix A. All RNA-seq data are available at the NCBI’s Gene Expression Omnibus under the accession PRJNA931465 (http://www.ncbi.nlm.nih.gov/bioproject/931465, accessed on 21 February 2023), aside from those reported in Appendix A of this paper.

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
