# Peer review of "Co-Regulatory Roles of WC1 and WC2 in Asexual Development and Photoreactivation of Beauveria bassiana"

_jof, 2023, doi:10.3390/jof9030290_

Round 1
Reviewer 1 Report
Scientific and statistical rigor is enough, evidence is well worked and explained to support conclusions adequately. It is suitable for publication as it is. Just a few comments:
line 40: recover from
line 167: (no. days) is not necessary, the units for LT50 is (d)
line 431: the teleomorphic form of Beauveria bassiana is Cordyceps bassiana.
Author Response
Please see attached a file for author response to review.

Reviewer 2 Report
The White Collar complex (WCC) is a well-studied blue-light photoreceptor and plays a central role in many fungi. As a light-responsive transcription factor complex, WCC is involved, for example, in the regulation of biosynthetic pathways of secondary metabolism and, in some species, in the control of the circadian clock. The heterodimeric WCC consist of White Collar 1 (WC1) and White Collar 2 (WC2). WC1 contains a specialized PAS domain (LOV domain), which harbors a flavin chromophore that enables perception of light. Both White Collar proteins typically possess zinc finger domains at each C-terminus to interact with DNA.
In the manuscript ‘Co-Regulatory Roles of WC1 and WC2 in Asexual Development 2 and Photoreactivation of Beauveria bassiana’ Xu et al. studied the role of WC1 and WC2 in the insect-pathogenic fungus B. bassiana using wc1/wc2 deletion mutants (DM) and complementation mutants (CM). DM and CM strains showed no phenotypic abnormalities compared to WT, but impaired conidiation rates and increased sensitivity to oxidative stress. Conidial survival rates after UVB treatment were significantly reduced in DM strains and rescued to WT level in CM strains. Microscopic analysis of germination and evaluation of conidia germination rates after UVB irradiation revealed the necessary role of WC1/WC2 in dark reactivation (NER) and photoreactivation (DNA photolyases). In addition, analysis of up-/down-regulated genes in DM strains compared to WT using RNAseq data confirmed the crucial role of WC1/WC2 in asexual reproduction of B. bassiana.
The experiments and analyses of this straightforward study sound thoroughly performed. The results are clear and of general importance for a better understanding of the role of the White Collar complex in general and in the insect-pathogenic fungus B. bassiana in particular. For the most part, the manuscript is well written. However, there are some typos, missing words and partly unclear parts that still need to be addressed (see below). Please check the entire manuscript carefully. Overall, I recommend publication in Journal of Fungi. I only have some minor criticisms and suggestions to consider:
Line 10: I assume… ‘WC1 and WC2’
Line 40-44: What exactly is meant by “shorter UV-induced DNA lesions” here? CPD lesions and (6-4) photoproducts are DNA lesions of two adjacent nucleobases (typically thymidines). Please specify.
Line 45: Please change to ‘DNA photolyase domain’ here and throughout the manuscript.
Line 48-49: What do you mean with ‘recent progress’? Please make a logical connection to the section.
Line 64-65: I suggest rephrasing the entire sentence.
Line 91-93: That sounds confusing/misleading to me. What exactly do you want to say with this sentence? WC1 and WC2 does typically not have a DNA photolyase domain. I therefore assume that neither WC1 nor WC2 has any direct light-dependent DNA repair activity (photorepair/photoreactivation). Please be more precise here or rephrase this sentence/section.
Line 98-100: This sentence is also confusing/misleading to me and my criticism is related to the point above (line 91-93). Please provide more detailed information.
Line 125: Missing word… ‘wc1’
Line 240-243: Missing or unnecessary word(s)… Please check the phrase ‘after a’.
Line 278: ‘impaired’
Line 293: ‘In the full-dark treatment, …’ (comma)
Line 338: Please rephrase this sentence.
Figure 2c/d: To make it more consistent, please change the symbol of ‘wc2 DM’ according to Figure 1f.
Author Response

(The authors gave the same response as above.)
